# Hiding Images in Plain Sight:
# Deep Steganography

**Shumeet Baluja**
Google Research
Google, Inc.
shumeet@google.com

## Abstract

Steganography is the practice of concealing a secret message within another, ordinary, message. Commonly, steganography is used to unobtrusively hide a small message within the noisy regions of a larger image. In this study, we attempt to place a full size color image within another image of the same size. Deep neural networks are simultaneously trained to create the hiding and revealing processes and are designed to specifically work as a pair. The system is trained on images drawn randomly from the ImageNet database, and works well on natural images from a wide variety of sources. Beyond demonstrating the successful application of deep learning to hiding images, we carefully examine how the result is achieved and explore extensions. Unlike many popular steganographic methods that encode the secret message within the least significant bits of the carrier image, our approach compresses and distributes the secret image's representation across all of the available bits.

## 1   Introduction to Steganography

*Steganography* is the art of covered or hidden writing; the term itself dates back to the 15th century, when messages were physically hidden. In modern steganography, the goal is to covertly communicate a digital message. The steganographic process places a hidden message in a transport medium, called the carrier. The carrier may be publicly visible. For added security, the hidden message can also be encrypted, thereby increasing the perceived randomness and decreasing the likelihood of content discovery even if the existence of the message detected. Good introductions to steganography and *steganalysis* (the process of discovering hidden messages) can be found in [1–5].

There are many well publicized nefarious applications of steganographic information hiding, such as planning and coordinating criminal activities through hidden messages in images posted on public sites – making the communication and the recipient difficult to discover [6]. Beyond the multitude of misuses, however, a common use case for steganographic methods is to embed authorship information, through digital watermarks, without compromising the integrity of the content or image.

The challenge of good steganography arises because embedding a message can alter the appearance and underlying statistics of the carrier. The amount of alteration depends on two factors: first, the amount of information that is to be hidden. A common use has been to hide textual messages in images. The amount of information that is hidden is measured in *bits-per-pixel (bpp)*. Often, the amount of information is set to 0.4bpp or lower. The longer the message, the larger the bpp, and therefore the more the carrier is altered [6, 7]. Second, the amount of alteration depends on the carrier image itself. Hiding information in the noisy, high-frequency filled, regions of an image yields less humanly detectable perturbations than hiding in the flat regions. Work on estimating how much information a carrier image can hide can be found in [8].

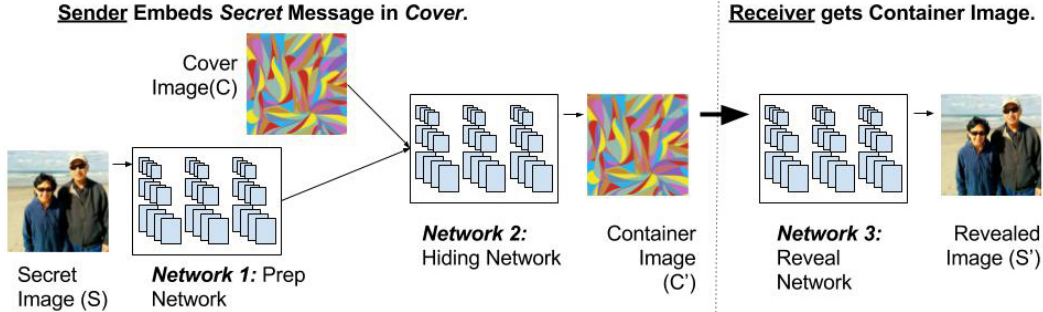

Figure 1: The three components of the full system. Left: Secret-Image preparation. Center: Hiding the image in the cover image. Right: Uncovering the hidden image with the reveal network; this is trained simultaneously, but is used by the receiver.

The most common steganography approaches manipulate the least significant bits (LSB) of images to place the secret information - whether done uniformly or adaptively, through simple replacement or through more advanced schemes [9, 10]. Though often not visually observable, statistical analysis of image and audio files can reveal whether the resultant files deviate from those that are unaltered. Advanced methods attempt to preserve the image statistics, by creating and matching models of the first and second order statistics of the set of possible cover images explicitly; one of the most popular is named HUGO [11]. HUGO is commonly employed with relatively small messages ($< 0.5bpp$). In contrast to the previous studies, we use a neural network to *implicitly model* the distribution of natural images as well as embed a much larger message, a full-size image, into a carrier image.

Despite recent impressive results achieved by incorporating deep neural networks with steganalysis [12–14], there have been relatively few attempts to incorporate neural networks into the hiding process itself [15–19]. Some of these studies have used deep neural networks (DNNs) to select which LSBs to replace in an image with the binary representation of a text message. Others have used DNNs to determine which bits to extract from the container images. In contrast, in our work, the neural network determines where to place the secret information and how to encode it efficiently; the hidden message is dispersed throughout the bits in the image. A decoder network, that has been simultaneously trained with the encoder, is used to reveal the secret image. Note that the networks are trained only once and are independent of the cover and secret images.

In this paper, the goal is to visually hide a full $N \times N \times RGB$ pixel secret image in another $N \times N \times RGB$ cover image, with minimal distortion to the cover image (each color channel is 8 bits). However, unlike previous studies, in which a hidden text message must be sent with perfect reconstruction, we relax the requirement that the secret image is losslessly received. Instead, we are willing to find acceptable trade-offs in the quality of the carrier and secret image (this will be described in the next section). We also provide brief discussions of the discoverability of the existence of the secret message. Previous studies have demonstrated that hidden message bit rates as low as 0.1bpp can be discovered; our bit rates are $10\times$ - $40\times$ higher. Though visually hard to detect, given the large amount of hidden information, we do not expect the existence of a secret message to be hidden from statistical analysis. Nonetheless, we will show that commonly used methods do not find it, and we give promising directions on how to trade-off the difficulty of existence-discovery with reconstruction quality, as required.

## 2   Architectures and Error Propagation

Though steganography is often conflated with cryptography, in our approach, the closest analogue is image compression through auto-encoding networks. The trained system must learn to compress the information from the secret image into the least noticeable portions of the cover image. The architecture of the proposed system is shown in Figure 1.

The three components shown in Figure 1 are trained as a single network; however, it is easiest to describe them individually. The leftmost, *Prep-Network*, prepares the secret image to be hidden. This component serves two purposes. First, in cases in which the secret-image (size $M \times M$) is smaller than the cover image ($N \times N$), the preparation network progressively increases the size of the secret image to the size of the cover, thereby distributing the secret image's bits across the entire $N \times N$

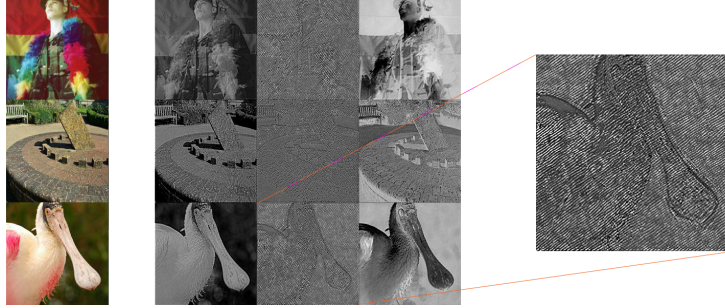

Figure 2: Transformations made by the preparation network (3 examples shown). Left: Original Color Images. Middle: the three channels of information extracted by the preparation network that are input into the middle network. Right: zoom of the edge-detectors. The three color channels are transformed by the preparation-network. In the most easily recognizable example, the 2nd channel activates for high frequency regions, e.g. textures and edges (shown enlarged (right)).

pixels. (For space reasons, we do not provide details of experiments with smaller images, and instead concentrate on full size images). The more important purpose, relevant to all sizes of hidden images, is to transform the color-based pixels to more useful features for succinctly encoding the image – such as edges [20, 21], as shown in Figure 2.

The second/main network, the *Hiding Network*, takes as input the output of the preparation-network and the cover image, and creates the *Container* image. The input to this network is a $N \times N$ pixel field, with depth concatenated RGB channels of the cover image and the transformed channels of the secret image. Over 30 architectures for this network were attempted for our study with varying number of hidden layers and convolution sizes; the best consisted of 5 convolution layers that had 50 filters each of $\{3 \times 3, 4 \times 4, 5 \times 5\}$ patches. Finally, the right-most network, the *Reveal Network*, is used by the receiver of the image; it is the decoder. It receives only the Container image (not the cover nor secret image). The decoder network removes the cover image to reveal the secret image.

As mentioned earlier, our approach borrows heavily from auto-encoding networks [22]; however, instead of simply encoding a single image through a bottleneck, we encode two images such that the intermediate representation (the container image) appears as similar as possible to the cover image. The system is trained by reducing the error shown below ($c$ and $s$ are the cover and secret images respectively, and $\beta$ is how to weigh their reconstruction errors):

$$\mathcal{L}(c, c', s, s') = ||\mathbf{c} - \mathbf{c}'|| + \beta ||\mathbf{s} - \mathbf{s}'|| \tag{1}$$

It is important to note where the errors are computed and the weights that each error affects, see Figure 3. In particular, note that the error term $||\mathbf{c} - \mathbf{c}'||$ does not apply to the weights of the reveal-network that receives the container image and extracts the secret image. On the other hand, all of the networks receive the error signal $\beta ||\mathbf{s} - \mathbf{s}'||$ for reconstructing the hidden image. This ensures that the representations formed early in the preparation network as well as those used for reconstruction of the cover image also encode information about the secret image.

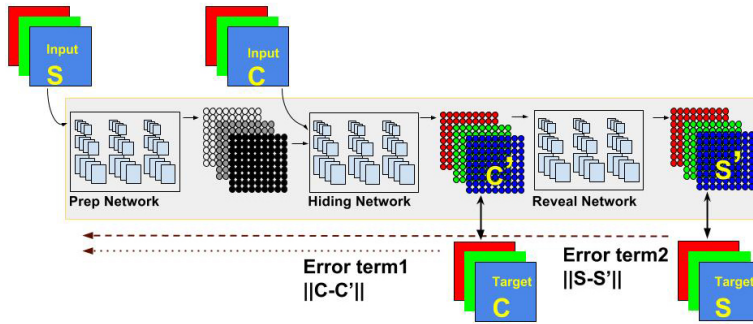

Figure 3: The three networks are trained as a single, large, network. Error term 1 affects only the first two networks. Error term 2 affects all 3. $S$ is the secret image, $C$ is the cover image.

To ensure that the networks do not simply encode the secret image in the LSBs, a small amount of noise is added to the output of the second network (*e.g.* into the generated container image) during training. The noise was designed such that the LSB was occasionally flipped; this ensured that the LSB was not the sole container of the secret image's reconstruction. Later, we will discuss where the secret image's information is placed. Next, we examine how the network performs in practice.

## 3   Empirical Evaluation

The three networks were trained as described above using Adam [23]. For simplicity, the reconstructions minimized the sum of squares error of the pixel difference, although other image metrics could have easily been substituted [24, 25]. The networks were trained using randomly selected pairs of images from the ImageNet training set [26].

Quantitative results are shown in Figure 4, as measured by the SSE per pixel, per channel. The testing was conducted on 1,000 image pairs taken from ImageNet images (not used in training). For comparison, also shown is the result of using the same network for only encoding the cover image without the secret image (e.g. $\beta = 0$). This gives the best reconstruction error of the cover using this network (this is unattainable while also encoding the secret image). Also shown in Figure 4 are histograms of errors for the cover and reconstruction. As can be seen, there are few large pixel errors.

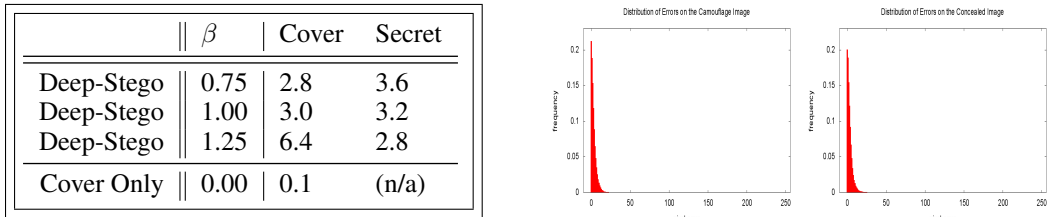

|  | $\beta$ | Cover | Secret |
|---|---|---|---|
| Deep-Stego | 0.75 | 2.8 | 3.6 |
| Deep-Stego | 1.00 | 3.0 | 3.2 |
| Deep-Stego | 1.25 | 6.4 | 2.8 |
| Cover Only | 0.00 | 0.1 | (n/a) |

Figure 4: Left: Number of intensity values off (out of 256) for each pixel, per channel, on cover and secret image. Right: Distribution of pixel errors for cover and secret images, respectively.

Figure 5 shows the results of hiding six images, chosen to show varying error rates. These images are *not* taken from ImageNet to demonstrate that the networks have not over-trained to characteristics of the ImageNet database, and work on a range of pictures taken with cell phone cameras and DSLRs. Note that most of the reconstructed cover images look almost identical to the original cover images, despite encoding all the information to reconstruct the secret image. The differences between the original and cover images are shown in the rightmost columns (magnified 5× in intensity).

Consider how these error rates compare to creating the container through simple LSB substitution: replacing the 4 least significant bits (LSB) of the cover image with the 4 most-significant 4-bits (MSB) of the secret image. In this procedure, to recreate the secret image, the MSBs are copied from the container image, and the remaining bits set to their average value across the training dataset. Doing this, the average pixel error per channel on the cover image's reconstruction is 5.4 (in a range of 0-255). The average error on the reconstruction of the secret image (when using the average value for the missing LSB bits) is approximately 4.0.[1] Why is the error for the cover image's reconstruction larger than 4.0? The higher error for the cover image's reconstruction reflects the fact that the distribution of bits in the natural images used are different for the MSBs and LSBs; therefore, even though the secret and cover image are drawn from the same distribution, when the MSB from the secret image are used in the place of the LSB, larger errors occur than simply using the average values of the LSBs. Most importantly, these error rates are significantly higher than those achieved by our system (Figure 4).

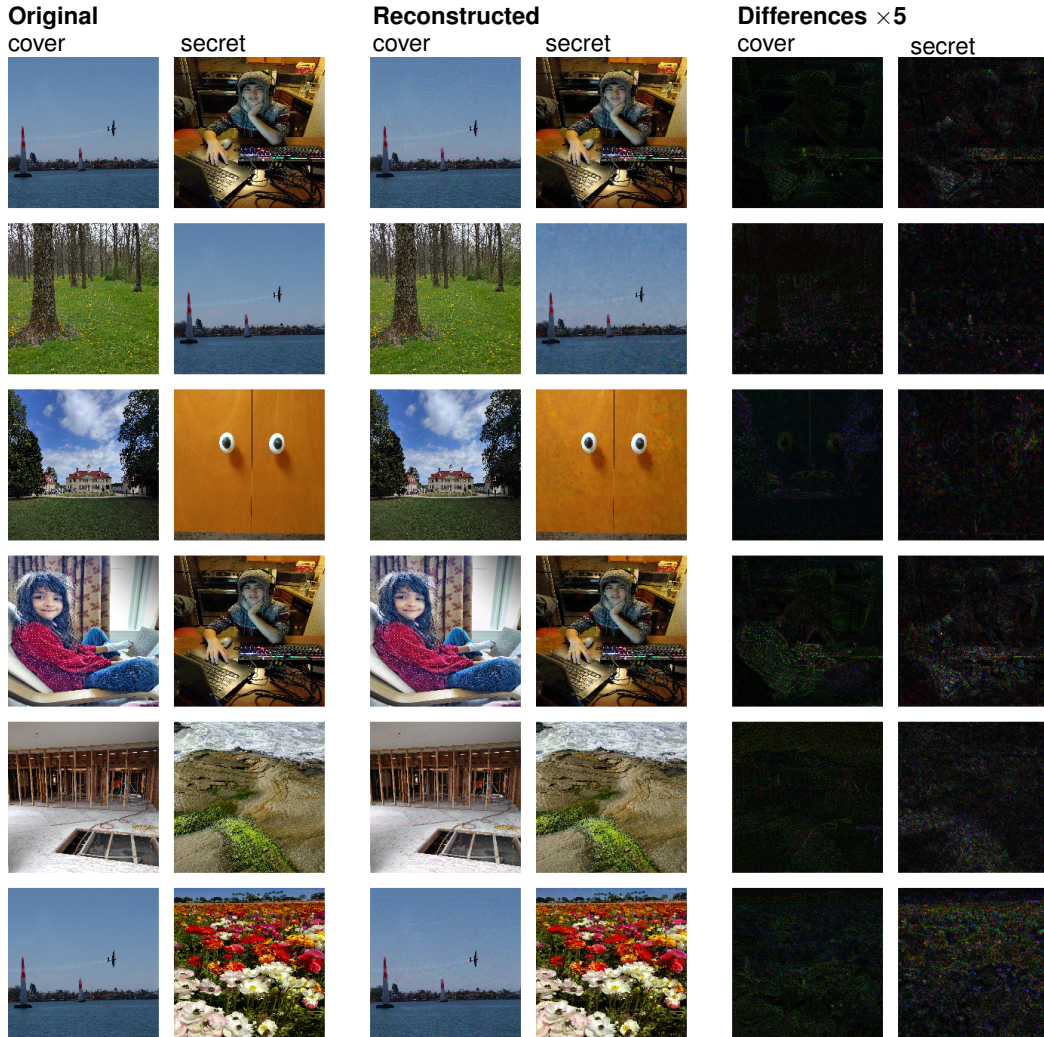

**Original**     **Reconstructed**     **Differences ×5**
cover    secret     cover    secret     cover    secret

Figure 5: 6 Hiding Results. Left pair of each set: original cover and secret image. Center pair: cover image embedded with the secret image, and the secret image after extraction from the container. Right pair: Residual errors for cover and hidden – enhanced 5×. The errors per pixel, per channel are the smallest in the top row: (3.1, 4.5) , and largest in the last (4.5, 7.9).

We close this section with a demonstration of the limitation of our approach. Recall that the networks were trained on natural images found in the ImageNet challenge. Though this covers a very large range of images, it is illuminating to examine the effects when other types of images are used. Five such images are shown in Figure 6. In the first row, a pure white image is used as the cover, to examine the visual effects of hiding a colorful secret image. This simple case was not encountered in training with ImageNet images. The second and third rows change the secret image to bright pink circles and uniform noise. As can be seen, even though the container image (4th column) contains only limited noise, the recovered secret image is extremely noisy. In the final two rows, the cover image is changed to circles, and uniform noise, respectively. As expected, the errors for the reconstruction of the cover and secret are now large, though the secret image remains recognizable.

### 3.1 What if the original cover image became accessible?

For many steganographic applications, it can safely be assumed that access to the original cover image (without the secret image embedded) is impossible for an attacker. However, what if the original cover image was discovered? What could then be ascertained about the secret image, even without access to the decoding network? In Figure 5, we showed the difference image between the original cover and the container with 5x enhancement – almost nothing was visible. We reexamine

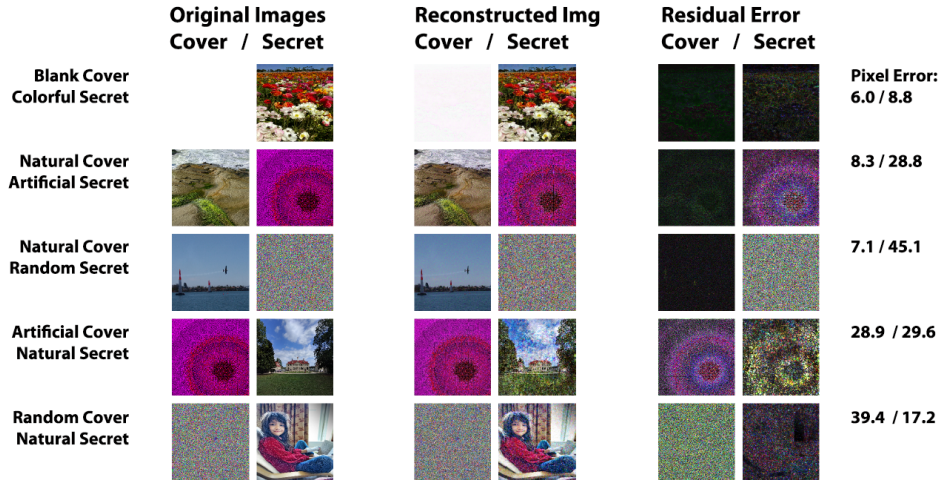

| | Original Images<br>Cover / Secret | Reconstructed Img<br>Cover / Secret | Residual Error<br>Cover / Secret | |
|---|---|---|---|---|
| Blank Cover<br>Colorful Secret | | | | Pixel Error:<br>6.0 / 8.8 |
| Natural Cover<br>Artificial Secret | | | | 8.3 / 28.8 |
| Natural Cover<br>Random Secret | | | | 7.1 / 45.1 |
| Artificial Cover<br>Natural Secret | | | | 28.9 / 29.6 |
| Random Cover<br>Natural Secret | | | | 39.4 / 17.2 |

Figure 6: Results with images outside the set of natural images.

the residual image at 5x, 10x, and 20x enhancement (with clipping at 255 where appropriate), see Figure 7. In the first row, note that the residual (at 20x) strongly resembles the cover image. In the second row, the residual is a combination of the cover and secret image, and in the third row, we see the most troubling result – features of the secret image are revealed. (Recall that this happens only when the original, unperturbed image, is available for comparison). There are many standard methods for obfuscation, such as adding cryptographic encodings of the secret image before embedding it into the cover image. We demonstrate another avenue that can be used in conjunction with any other approach: modifying the network's error function.

In addition to the two error terms described, we add an error term that minimizes the pixel-wise correlation between the *residual of the cover image* and the secret image $corr(R_c, S)$ where $R_c = ||C - C'||$ and $S$ is the secret image. Many weightings for this term were empirically tested. In the results shown in Figure 7(Bottom), it is scaled to approximately (0.15 * number of pixel * channels). Minimizing the residual's correlation with the secret image removed many of the secret image's features from the residuals – even when 20x magnification is used. Naturally, the robustness and resilience comes at a price; the quality of some of the reconstructions have decreased, as shown (see saturation of reds in first image).

## 4 Where is the Secret Image Encoded?

The primary focus of this paper is to concretely demonstrate that it is possible to encode a large amount of information in an image with limited visually noticeable artifacts. However, no explicit attempt has been made to actively hide the existence of that information from machine detection. Though we cannot expect to completely hide the fact that up to ½ of the information is part of a hidden message, measures can be taken to make it more difficult to discover. First, however, we must determine where the information of the secret image resides.

Is the network simply hiding the information about the secret image in the least significant bits of the cover image? Tools exist to seek out hidden information in the LSBs. One such publicly available steganalysis toolkit, **StegExpose**, was used to test the detectability of our hidden images [27–29]. Per the description of the tool: *"StegExpose rating algorithm is derived from an intelligent and thoroughly tested combination of pre-existing pixel based steganalysis methods including Sample Pairs by Dumitrescu (2003), RS Analysis by Fridrich (2001), Chi Square Attack by Westfeld (2000) and Primary Sets by Dumitrescu (2002)"* [27]. In addition to the default settings (threshold = 0.2), the detection thresholds were varied throughout a large range. The ROC curve for StegExpose is shown in Figure 8. Note the little variation beyond random guessing (the green line).

StegExpose should have been able to find the information if it were simply placed in the LSB bits. We turn to a second method to find where the information is stored. The images used in the study

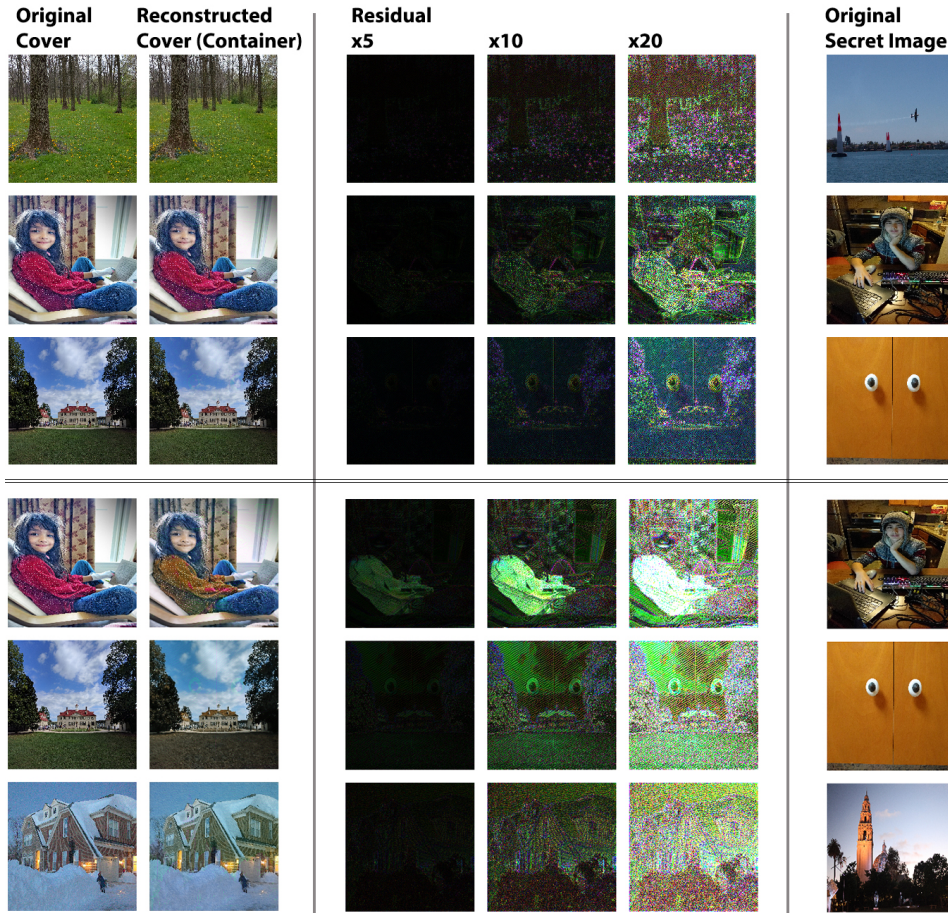

| Original Cover | Reconstructed Cover (Container) | Residual x5 | x10 | x20 | Original Secret Image |

Figure 7: Top 3 rows. If the original image is leaked and is subtracted from the container image, the residual can be computed. With enough enhancement (20x), some of the secret image is revealed. Bottom 3 rows: by explicitly creating an error term that minimized the correlation between the residual and the secret image, the residual reveals less about the secret image; however, the pixel errors for the container rise (note the less saturated colors in some of the red regions).

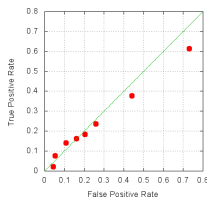

Figure 8: ROC curves: True Positive Rate vs. False Positive Rate for **StegExpose** when trying to detect images embedded via the proposed method.

are composed, at each pixel, of 24 bits ($8 \times (R, G, B)$). If we flip the first bit of the R channel of all the pixels in the *container image*, we can measure its effects on the reconstructions on the container image itself and also, by propagating the modified image through reveal network, on the reconstruction of the secret image. The effects are striking, see Figure 9.

In Figure 9, the left half of the figure should be considered the "control". The upper left bar-chart, shows that the effects of changing a bit in the Red channel of the container only has an effect on the red-channel in the container, and that the magnitude is proportional to the significance of the bit; this is exactly as expected and holds true for the Green and Blue channels as well (shown on the diagonal). Much more interesting is the right side of Figure 9. Here, we see that a bit flip in any bit position in any color channel of the **container image**, has an effect across *all color channels* in the secret

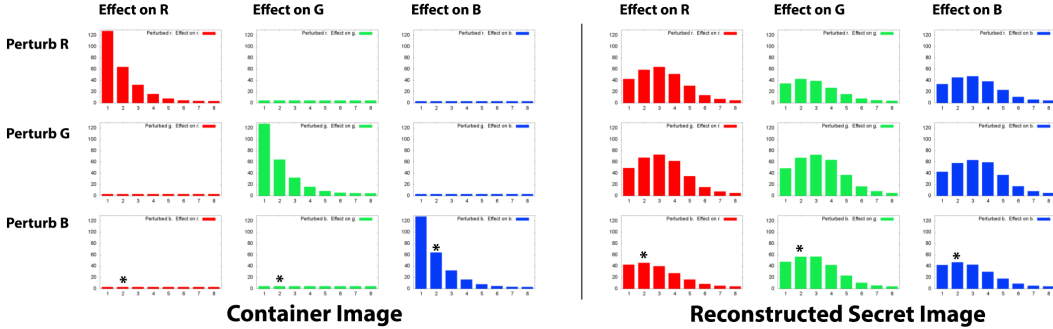

Figure 9: Sensitivity to Bits in the Container Image. (Left:) Control case, the container image. Changing a bit in the container image obviously has no effect on other bits. (Right): However, that same bit flip in the container image has effects across all color bands in the recovered secret image. For example, if we want to measure the effect of flipping the 2nd bit in the Blue channel, we would look at the marked entries (marked with *).

image's reconstruction. Moreover, the error does not follow the standard significance ordering of the bit positions. In other words, the information for the secret image is spread across the color channels; hence the reason it was not detected earlier.

Despite StegExpose not finding the encodings, we suspected that this amount of information should be readily discoverable. It has recently been demonstrated in multiple studies that deep networks are competitive and often better than the more established methods of steganalysis that use hand selected image features [12–14]. Interestingly, many published steganalysis techniques are *not blind*: they assume that they are detecting hidden images only from known hiding methods and even that access to the statistics about the cover image distribution are available. This makes the steganalysis task easier while making the hiding task much more difficult. We will not discuss the real-world limitations of that assumption here, but will follow that example.

We trained the steganalysis networks as binary classifiers, using the unperturbed ImageNet images as negative samples, and our containers as positive examples. We discovered that the steganalysis networks performed well with $24 \times (N \times N)$ binary input layers. Each of the 24 layers represented a bit in the pixel representation of the input image. The networks were trained in an analogous manner to others in this study; again a multitude of architectures and learning rates were attempted.

The fully convolutional networks were able to obtain between 90-95% combined classification rates. For comparison, the same networks were retrained to find Least-Significant-Bit substitution (where the $L$ least significant bits of each color channel in the cover image are replaced with the $L$ most significant bits from the secret image). The networks were trained and tested with $L = 1, 2, 3, 4$; all achieved over 99% combined rates. An additional set of tests revealed that even when a single bit was randomly placed in one of the 4 bit positions, the detection rate remained over 99%. Given these detection rates on even a single bit, the high detection rates with a full image are not surprising.

Nonetheless, this leads to an immediate direction for future work. In this study, the steganalysis networks were trained and used after the complete encoding system was created. However, using pre-trained and/or simultaneously trained steganalysis networks in an adversarial learning framework, such as Generative-Adversarial-Networks (GAN) [30], during the training of steganography nets provides a method to incorporate an obfuscation based error metric. The adversary provides a supplemental error signal based upon the discoverability of the hidden message that is minimized in addition to the reconstruction errors. [31] has recently successfully explored a very similar procedure with small messages hidden in images.

## 5   Discussion & Future Work

In this section, we briefly discuss a few observations found in this study and present ideas for future work. First, lets consider the possibility of training a network to recover the hidden images *after* the system has been deployed and without access to the original network. One can imagine that if an

attacker was able to obtain numerous instances of container images that were created by the targeted system, and in each instance if at least one of the two component images (cover or secret image) was also given, a network could be trained to recover both constituent components. What can an attacker do without having access to this ground-truth "training" data? Using a smoothness constraint or other common heuristic from more classic image decomposition and blind source separation [32–34] may be a first alternative. With many of these approaches, obtaining even a modest amount of training data would be useful in tuning and setting parameters and priors. If such an attack is expected, it is open to further research how much adapting the techniques described in Section 3.1 may mitigate the effectiveness of these attempts.

As described in the previous section, in its current form, the correct detection of the existence (not necessarily the exact content) of a hidden image is indeed possible. The discovery rate is high because of the amount of information hidden compared to the cover image's data (1:1 ratio). This is far more than state-of-the-art systems that transmit reliably undetected messages. We presented one of many methods to make it more difficult to recover the contents of the hidden image by explicitly reducing the similarity of the cover image's residual to the hidden image. Though beyond the scope of this paper, we can make the system substantially more resilient by supplementing the presented mechanisms as follows. Before hiding the secret image, the pixels are permuted (in-place) in one of $M$ previously agreed upon ways. The permuted-secret-image is then hidden by the system, as is the key (an index into $M$). This makes recovery difficult even by looking at the residuals (assuming access to the original image is available) since the residuals have no spatial structure. The use of this approach must be balanced with (1) the need to send a permutation key (though this can be sent reliably in only a few bytes), and (2) the fact that the permuted-secret-image is substantially more difficult to encode; thereby potentially increasing the reconstruction-errors throughout the system. Finally, it should be noted that in order to employ this approach, the trained networks in this study *cannot* be used without retraining. The entire system must be retrained as the hiding networks can no longer exploit local structure in the secret image for encoding information.

This study opens a new avenue for exploration with steganography and, more generally, in placing supplementary information in images. Several previous methods have attempted to use neural networks to either augment or replace a small portion of an image-hiding system. We have demonstrated a method to create a fully trainable system that provides visually excellent results in unobtrusively placing a full-size, color image into another image. Although the system has been described in the context of images, the same system can be trained for embedding text, different-sized images, or audio. Additionally, by using spectrograms of audio-files as images, the techniques described here can readily be used on audio samples.

There are many immediate and long-term avenues for expanding this work. Three of the most immediate are listed here. (1) To make a complete steganographic system, hiding the existence of the message from statistical analyzers should be addressed. This will likely necessitate a new objective in training (*e.g.* an adversary), as well as, perhaps, encoding smaller images within large cover images. (2) The proposed embeddings described in this paper are not intended for use with lossy image files. If lossy encodings, such as jpeg, are required, then working directly with the DCT coefficients instead of the spatial domain is possible [35]. (3) For simplicity, we used a straightforward SSE error metric for training the networks; however, error metrics more closely associated with human vision, such as SSIM [24], can be easily substituted.

## Footnotes

[1]Note that an error of 4.0 is expected when the average value is used to fill in the LSB: removing 4 bits from a pixel's encoding yields 16x fewer intensities that can be represented. By selecting the average value to replace the missing bits, the maximum error can be 8, and the average error is 4, assuming uniformly distributed bits. To avoid any confusion, we point out that though it is tempting to consider using the average value for the cover image also, recall that the LSBs of the cover image are where the MSBs of the secret image are stored. Therefore, those bits must be used in this encoding scheme, and hence the larger error.

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
