[Reviews · NeurIPS 2017]

Reviewer 1



The authors present a new steganography technique based on deep neural networks to simultaneously conduct hiding and revealing as a pair. The main idea is to combine two images of the same size together. The trained process aims to compress the information from the secret image into the least noticeable portions of the cover image and consists of three processes: a prep-Network for encoding features, the Hiding Network creates a container image, and a Reveal Network for decoding the transmitted container image. On the positive side, the proposed technique seems novel and clever, although it uses/modifies existing deep learning frameworks and therefore should be viewed as an application paper. The experiments are comprehensive and the results are convincing. The technique, however, resembles greatly the image decomposition problem, e.g., for separating mixtures of the intrinsic and reflection layers in previous literatures. I'd hope the authors to clarify how the problems are different and if the proposed technique can be used to resolve the layer separation problem. More important, I wonder if existing approaches on layer separations can be used to directly decode your encrypted results. I am also a bit concerned about the practicality of the proposed technique. First, since the container image will be transmitted and potentially intercepted. It appears that one can tell directly that the container image contains hidden information (the image contains ringing type visual artifacts). if that is the case, the approach is likely to undermine the effort. Second, if the cover and secret images appear similar, the technique may fail to robustly separate them. So a more interesting question is how to pick a suitable cover image for a specific secret image. But such discussions seem largely missing. Third, the requirement the cover and the secrete images should have the same size seems to be a major limitation. One would have to resize the images to make them match, it may increase file sizes, etc. Overall, I think the proposed approach is interesting but I have concerns on the practicality and would like to see comparisons with state-of-the-art layer separation techniques.

Reviewer 2



The authors propose a method of using deep neural networks to embed a secret image unobtrusively into a carrier image so that it can be transmitted without drawing attention to it. The proposed architecture shares some ideas with auto-encoders. The network is trained with a joint objective so that the carrier image with the embedded image is similar to the original cover and the reconstructed secret image is similar to the original secret image. This is a lossy encoding so it is not reconstructed perfectly which may be acceptable for images that won't be subject to forensics. One question that remained for me is the degree to which the reconstructed image is a domain based reconstruction ... is it filling the image from understanding of the domain characteristics? The authors anticipate many questions that occur to the reader such as how does this compare to least significant bit encoding, how is the embedding stored and how does it perform on images not in the training set. For instance, the authors show that typical LSB-based steganography detection methods fail on their encoded images. The authors also investigate detectability and possible exploits such as using the residual between the original cover image and the encoded image to try and gain information about the secret. They show that adding a penalty for correlation between this residual and the original secret reduces the informativeness of the residuals. The practical applicability, advances over state of the art in terms of both coding density and detectability combined with the insightful analysis about how the embedding is stored make this paper both practically significant and theoretically enlightening (though there are no formal theorems in the paper). It is essentially an applied paper that uses existing techniques in a novel way.

Reviewer 3



This paper tries to hide one full size color image into another of the same size (steganography) using deep neural networks. The framework is relatively simple, and results look encouraging. I have the following major comment: The author should compare the proposed steganography framework with existing steganography techniques. I understand that existing steganography techniques normally only hide a small message within the noisy regions of a larger image, but the authors should at least show some comparison on those tasks. Other than that, the frameworks looks simple and clean, and I do think that it could be the new avenue for exploration on steganography.